# Spatiotemporal Change and Coordinated Development Analysis of "Population-Society-Economy-Resource-Ecology-Environment" in the Jing-Jin-Ji Urban Agglomeration from 2000 to 2015

**Jianwan Ji [1,2]** , **Shixin Wang [1,\*]**, **Yi Zhou [1]**, **Wenliang Liu [1,\*]** and **Litao Wang [1]**

[1]   Aerospace Information Research Institute, Chinese Academy of Sciences, Beijing 100094, China; jijw@radi.ac.cn (J.J.); zhouyi@radi.ac.cn (Y.Z.); wanglt@radi.ac.cn (L.W.)
[2]   University of Chinese Academy of Sciences, Beijing 100049, China
\*   Correspondence: wangsx@radi.ac.cn (S.W.); liuwl@radi.ac.cn (W.L.)

**Abstract:** Measuring the regionally coordinated development degree quantitively at an urban agglomeration scale is vital for regional sustainable development. To date, existing studies mainly utilized statistical data to analyze coordinated development degrees between different subsystems, which failed to measure the development gap of subsystems between cities. This study integrated remote sensing and statistical data to evaluate the development degree from six subsystems. The coordinated index (CI) and coordinated development index (CDI) were then promoted to assess the coordinated degree and coordinated development degree. The main findings were: (1) The coordinated development degree of Jing-Jin-Ji (JJJ) had increased from 0.4616 in 2000 to 0.6099 in 2015, with the corresponding grade improvement from "moderate" to "good"; (2) JJJ and six subsystems' development degree showed an increasing trend. JJJ's whole development degree had improved from 0.34 to 0.52, and the grade had changed from "fair" to "moderate"; (3) The coordinated degree of JJJ displayed a "V" shape. However, the coordinated degree was lower in 2015 than in 2000.

**Keywords:** coordinated development degree; Jing-Jin-Ji; remote sensing; complex system



## 1. Introduction

Since the implementation of the reform and opening-up policy in 1978, tremendous changes have occurred in mainland China [1,2]. However, accompanied by a high economic growth speed, several problems have arisen, such as a low resource utilization efficiency, a high degree of environmental pollution, a wider economic development gap, etc. In response to these problems, the Chinese government promulgated the regional coordinated development strategy in 2017. To date, numerous studies have focused on this topic. Specifically, from the point of view of definition, it could be divided into two types. The first one refers to the synchronized development of different subsystems. These subsystems included population, society, economy, environment, etc. For instance, Li and Yi assessed the city's coordinated development level from three subsystems, which were economy, society, and environment [3]; Xie et al. analyzed the coordinated development of the "resources-environment-ecology-economy-society" complex system in China [4]. The second one refers to the economic gap narrowing between different regions. For example, Qin et al. evaluated China's economic gap and its coordinated development level [5]. However, both definitions consider one-sided aspects of development. The first one emphasized the synchronism of various subsystems but failed to assess each subsystem's gap between regions. The second one only measured the regional economic gap while ignoring other subsystems, such as population, society, resources, etc. From a methodological point of view, these studies can be grouped into three types: the development degree model, the coordinated degree model, and the coordinated development degree model. The first type

includes the principal component analysis (PCA); for instance, Bolcárová and Košta [6]. The second type includes grey relational analysis [7], data envelopment analysis [8], system dynamic analysis [9], etc. The third type includes the coupling coordinated degree model (CCD) [10,11]. For example, Lin et al. evaluated the coupling coordination changes between the urbanization quality and eco-environment pressure of the West Taiwan Strait urban agglomeration [12]. Li et al. assessed the coordinated development between social economy and ecological environment in northeastern China [13]. Fan et al. studied the coupling coordinated development situation between social economy and ecological environment in Chinese provincial capital cities [14]. In general, these studies measured mainly the coordinated development degree under the first definition. As for data sources, traditional research mainly applied statistical data to evaluate the coordinated development degree. For instance, Zameer et al. investigated the coordinated development of natural resources, financial development, and ecological efficiency in China [15]; Zhang et al. used panel data to assess the development level of the "Five Modernizations" [16]; Ma et al. utilized statistical data to evaluate the coordinated development from the perspective of new urbanization [17]. Generally, statistical data have their advantages, like high authority and accuracy; however, they also have some disadvantages, such as the inconsistent statistical caliber, limited coverage, and incomplete statistical indicators. Compared with statistical data, non-statistical data have been gradually utilized to investigate the regionally coordinated development degree. For example, Ariken et al. evaluated the coordinated development degree of urbanization and eco-environment in Yanqi Basin based on multi-source remote sensing data [18]; Tian et al. combined land use data and statistical data to analyze the coordination state between urbanization and ecosystem services [19]; Yang et al. integrated spatial data, environmental data, and statistical data to evaluate the coupling coordination of geo-ecological environment and urbanization [20]; Shao et al. explored the relationship between urbanization and ecological environment using remote sensing images and statistical data [21]. Based on these studies, we found that remote sensing data were gradually applied to evaluate the ecological quality. However, due to the ecology subsystem's complexity, numerous studies mainly assessed one aspect of it [22]. In this context, one comprehensive index, named remote sensing ecological index (RSEI), was promoted by Xu in 2013 [23]. Since its promotion, this index has been widely applied in numerous studies; however, it was seldom applied to assess the ecological quality at a large scale due to some limitations, like the difficulty of obtaining the same period's data and cloud pollution [24]. Integrated with MODIS datasets and Google Earth Engine technologies, Ji et al. successfully evaluated the regional ecological quality at an urban agglomeration scale [24,25].

As for the weighting methods, existing studies could be divided into subjective weighting methods, such as the analytic hierarchy process (AHP) [4,26,27] and the Delphi method [28], objective weighting methods, such as global principal component analysis [29] and the entropy method (EM) [30–35], and combined weighting methods, such as the fuzzy analytical hierarchy process (FAHP) and the entropy method [36,37]. In general, subjective weighting methods have the advantage of considering the expert's experience while ignoring the information of the data themselves. On the contrary, objective weighting methods consider the data's information while ignoring the decision makers' subjective intentions. Combined weighting methods utilize the advantages of both subjective and objective weighting methods, which have become more and more popular. Regarding the subsystems' selection, we concluded that population, economy, society, resource, ecology, and environment were the primary subsystems [38–43]. However, due to the similarity between environmental and ecological concepts, both subsystems' selected indicators were also similar [11,36]. The concept of environment mainly refers to the interaction between human beings and their living environment, while ecology mainly refers to the natural ecological situation.

Based on the above literature review, we first defined the regional coordinated development as "Within a certain geographical space, the internal development level of its

population, society, economy, resource, ecology, as well as environment tend to improve, and the internal differences tend to narrow, thus moving forward as a whole." Then, we constructed the indicator system from six subsystems: population, society, economy, resource, ecology, and environment, and assigned each subsystem's weight by combining AHP and EM. Finally, we introduced one novel coordinated index (CI) and calculated the coordinated development index (CDI) to analyze Jing-Jin-Ji's (JJJ) changes during 2000–2015. This paper is organized into five sections. The first section introduces the background of the study and its rationale. The material and methods section introduces the study area, the data source and indicator calculation, the acquisition of combined weighting, and the CI and CDI calculation. The result section incorporates the development degree and coordinated degree of JJJ and each subsystem in different years. The discussion part explains the weight allocation result, cause analysis, implications, and limitations of the study. The last section represents the conclusion of the study.

## 2. Materials and Methods

### 2.1. Study Area

The JJJ urban agglomeration is located in northern China, covering approximately 218,000 km² (Figure 1). It has a total of 13 cities with different categories. Beijing (BJ) and Tianjin (TJ) are municipalities, while Shijiazhuang (SJZ), Tangshan (TS), Qinhuangdao (QHD), Handan (HD), Xingtai (XT), Baoding (BD), Zhangjiakou (ZJK), Chengde (CD), Cangzhou (CZ), Langfang (LF), and Hengshui (HS) are prefecture-level cities. As one of China's three economic growth engines in the 21st century, JJJ's further development could contribute a lot to China's growth. In 2019, the total population and GDP reached 113.07 million people and 8458.00 billion yuan, accounting for 8.10% and 8.54% of the country's overall values.

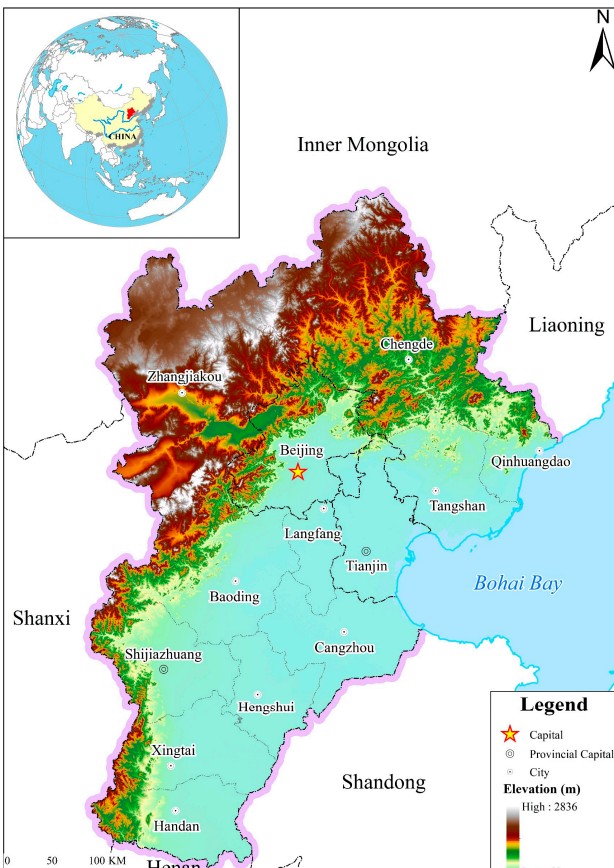

**Figure 1.** Location of the study area.

## 2.2. Indicator System and Data Collection

Based on the literature [3–21,26–43], we have collected 232 indicators at the beginning. Then, correlation analyses were done to exclude indicators with a high correlation coefficient. Finally, 25 indicators were selected based on the principles of comparability, accessibility, representativeness, and computability (Table 1).

**Table 1.** Indicator system.

| Subsystems | Attributes | Indicators | Unit | Trend | Data Source |
|---|---|---|---|---|---|
| Population | Population quantity | Total resident population ($C_1$) | 10,000 person | Negative | (1)(2)(3)(4) |
| | Population structure | Proportion of employees in the tertiary industry ($C_2$) | % | Positive | (1)(2)(3)(4) |
| | | Proportion of the urban population ($C_3$) | % | Positive | (1)(2)(3)(4) |
| | Education level | Per capita of years of education ($C_4$) | Year | Positive | (2)(3)(4)(5)(6) |
| Society | Infrastructure construction level | Number of mobile phone users per 100 people ($C_5$) | Users | Positive | (1)(2)(3)(4) |
| | | Per capita built-up area ($C_6$) | m$^2$ | Positive | (1)(2)(3)(4) |
| | | Road density ($C_7$) | m/km$^2$ | Positive | (7) |
| | Basic public service level | Number of doctors per 10,000 people ($C_8$) | People | Positive | (1)(2)(3)(4) |
| | | Number of beds in hospitals and health centers per 10,000 people ($C_9$) | Unit | Positive | (1)(2)(3)(4) |
| | | Number of buses and trams per 10,000 people ($C_{10}$) | Unit | Positive | (1)(2)(3)(4) |
| | | Number of full-time teachers in universities per 10,000 people ($C_{11}$) | Person | Positive | (1)(2)(3)(4) |
| | | Number of books in public libraries per 100 people ($C_{12}$) | Unit | Positive | (1)(2)(3)(4) |
| | | Proportion of basic medical insurance ($C_{13}$) | % | Positive | (1)(2)(3)(4) |
| Economy | Economic quantity | GDP ($C_{14}$) | Billion yuan | Positive | (1)(2)(3)(4) |
| | | Per capita GDP ($C_{15}$) | Yuan | Positive | (1)(2)(3)(4) |
| | Economic structure | Proportion of tertiary industry in GDP ($C_{16}$) | % | Positive | (1)(2)(3)(4) |
| Resource | Resource utilization efficiency | Water consumption per 10,000 yuan GDP ($C_{17}$) | Ton | Negative | (1)(2)(3)(4) |
| | | Energy consumption per 10,000 yuan GDP ($C_{18}$) | Ton's standard coal | Negative | (1)(2)(3)(4) |
| | | Comprehensive utilization rate of industrial solid waste ($C_{19}$) | % | Positive | (1)(2)(3)(4) |
| Ecology | Ecological quality | Remote sensing ecological index ($C_{20}$) | Non-dimension | Positive | (8)(9) |
| Environment | Environmental pollution level | PM$_{2.5}$ annual average concentration ($C_{21}$) | ug/m$^3$ | Negative | (10) |
| | | Per capita industrial wastewater discharge ($C_{22}$) | Ton | Negative | (1)(2)(3)(4) |

**Table 1.** Cont.

| Subsystems | Attributes | Indicators | Unit | Trend | Data Source |
|---|---|---|---|---|---|
| | | Industrial sulfur dioxide emissions per 10,000 people ($C_{23}$) | Ton | Negative | (1)(2)(3)(4) |
| | Environmental governance level | Harmless treatment rate of domestic garbage ($C_{24}$) | % | Positive | (1)(2)(3)(4) |
| | | Domestic sewage treatment rate ($C_{25}$) | % | Positive | (1)(2)(3)(4) |

In Table 1, the data sources are: (1) Chinese City Statistical Yearbook (2001, 2006, 2011, 2016); (2) Beijing Statistical Yearbook (2001, 2006, 2011, 2016); (3) Tianjin Statistical Yearbook (2001, 2006, 2011, 2016); (4) Hebei Economic Yearbook (2001, 2006, 2011, 2016); (5) Population Census of Hebei Province (2000, 2010); (6) Population Census of China (2000, 2010); (7) National Traffic Digital Map in vector format (2000, 2007, 2010, 2015); (8) MOD09A1 dataset (2001, 2005, 2010, 2015); (9) MOD11A2 dataset (2001, 2005, 2010, 2015); (10) PM$_{2.5}$ dataset (2000, 2005, 2010, 2015) [44]. For 8 to 10, the details are shown in Table 2.

**Table 2.** Introduction of the remote sensing dataset.

| Name | Spatial Resolution/m | Temporal Resolution/Day | Data Availability |
|---|---|---|---|
| MOD09A1 | 500 | 8 | https://lpdaac.usgs.gov/products/mod09a1v006/ accessed on 15 June 2019 |
| MOD11A2 | 1000 | 8 | https://lpdaac.usgs.gov/products/mod11a2v006/ accessed on 20 June 2019 |
| PM$_{2.5}$ | 1000 | Monthly | http://doi.org/10.5281/zenodo.3987359 accessed on 10 November 2020 |

Based on Table 2, the MOD09A1 product provided an estimate of the surface spectral reflectance of Terra MODIS bands 1–7 at 500 m resolution and corrected for atmospheric conditions. The MOD11A2 product provides an average 8-day land surface temperature (LST) at 1000 m resolution [45]. To exclude the phenology influence on RSEI [22,23], each year's time period was from 1 June to 31 October. Equations (1)–(6) were used to acquire JJJ's RSEI [22,23]. These equations were computed with the help of the Google Earth Engine platform and ArcGIS 10.6 software.

$$Greeness = NDVI = (\rho_{nir} - \rho_{red})/(\rho_{nir} + \rho_{red}) \tag{1}$$

$$WET = 0.1084 \times \rho_{red} + 0.0912 \times \rho_{nir} + 0.5065 \times \rho_{blue} + 0.4040 \times \rho_{green} - 0.2410 \times \rho_{mir1} - 0.4658 \times \rho_{mir2} - 0.5306 \times \rho_{mir3} \tag{2}$$

$$Dryness = NDBSI = \frac{1}{2}\left\{\frac{\frac{2\times\rho_{mir2}}{\rho_{mir2}+\rho_{nir}} - \frac{\rho_{nir}}{\rho_{nir}+\rho_{red}} - \frac{\rho_{green}}{\rho_{green}+\rho_{mir2}}}{\frac{2\times\rho_{mir2}}{\rho_{mir2}+\rho_{nir}} + \frac{\rho_{nir}}{\rho_{nir}+\rho_{red}} + \frac{\rho_{green}}{\rho_{green}+\rho_{mir2}}}\right\} + \frac{1}{2}\left\{\frac{\rho_{mir2}+\rho_{red}-\rho_{nir}-\rho_{blue}}{\rho_{mir2}+\rho_{red}+\rho_{nir}+\rho_{blue}}\right\} \tag{3}$$

$$RSEI_{origin} = 1 - PC1\{f(Greeness, Wet, Dryness, LST)\} \tag{4}$$

$$X_{rescale} = (X_i - X_{\min})/(X_{\max} - X_{\min}) \tag{5}$$

$$RSEI = (RSEI_{origin\_i} - RSEI_{\min})/(RSEI_{\max} - RSEI_{\min}) \tag{6}$$

where $\rho$ stands for the surface reflectance bands; *blue, green, red, nir, mir1, mir2, mir3* are the MODIS bands at 459–479 nm, 545–565 nm, 620–670 nm, 841–876 nm, 1230–1250 nm, 1628–1652 nm, and 2105–2155 nm, respectively; PC1 represents the first component of the spatial principal component analysis (SPCA); LST denotes the land surface temperature;

$X_{rescale}$ indicates the normalized result of each index; and RSEI means the normalized result of the $RSEI_{orgin}$.

### 2.3. The Indicator System Consistency Test and Weight Assignment

After constructing the indicator system, Cronbach's alpha test was adopted to test its consistency [46]. Equation (7) was the calculation formula.

$$\alpha = \frac{k\bar{r}}{1 + (k-1)\bar{r}} \tag{7}$$

where $\alpha$ is Cronbach's alpha, $k$ is the number of selected indicators, and $\bar{r}$ is the average value of all indicator's correlation coefficients. The $\alpha$ ranges from 0 to 1. This range can be further categorized into >0.9, 0.8–0.9, 0.7–0.8, and <0.7, that respectively reveals the indicator system is good, acceptable, has certain problems, and is not acceptable.

All the $\alpha$ test result values were greater than 0.87, and the average value was 0.8977 (Table 3), which was close to the threshold value (0.9), indicating that the indicator system was acceptable.

**Table 3.** The Cronbach's alpha values of the indicator system.

| Indicator System | Cronbach's Alpha Value |
|:---:|:---:|
| 2000 | 0.9256 |
| 2005 | 0.8992 |
| 2010 | 0.8770 |
| 2015 | 0.8887 |

Concerning weight assignment, equal weight was given to the indicators under the same subsystem. Each subsystem was given appropriate weight by combining AHP and EM. Before this, all indicators were rescaled to the same dimension. There are several normalization methods, such as the max-min, Z-score, variation coefficient, the ideal value, etc. Of these, we utilized the ideal value normalized method, because it allows for the comparison of the results of different years. Table 4 shows the ideal value and original value range of each indicator in 2000, 2005, 2010, and 2015. Equations (8) and (9) were the normalization formulas.

$$N_{ij} = \frac{x_{ij}}{x_{ideal}}, \; if \; it \; is \; a \; positive \; indicator, \tag{8}$$

$$N_{ij} = \frac{x_{ideal}}{x_{ij}}, \; if \; it \; is \; a \; negative \; indicator, \tag{9}$$

where $N_{ij}$ indicates the $i$th indicator's normalized value in the $j$th city; $x_{ideal}$ represents the $i$th indicator's ideal value.

After normalization, all indicators' original values were transformed into 0–1. Each subsystem's development degree value (DI) was computed using Equation (10).

$$DI = \sum_{i=1}^{m} w_i N_{ij} \tag{10}$$

where DI indicates a subsystem's development degree value, from 0 to 1; $N_{ij}$ is the normalized value of each indicator; $w_i$ denotes each indicator's weight within the same subsystem. Take the resource subsystem as an example, it has three indicators ($C_{17}$, $C_{18}$, $C_{19}$). They were assigned the same weight (1/3). Then, the development degree value of the resource subsystem was acquired based on Equation (10).

**Table 4.** The ideal value of each indicator.

| Indicator | Ideal Value | Data Range |
|---|---|---|
| Total resident population ($C_1$) | Highest population carrying capacity value [47] | 275.40–2170.50 |
| Proportion of employees in the tertiary industry ($C_2$) | Highest value of all years | 39.80–80.07 |
| Proportion of the urban population ($C_3$) | 70% [48] | 15.87–86.51 |
| Per capita of years of education ($C_4$) | 15 years [49] | 7.43–12.65 |
| Number of mobile phone users per 100 people ($C_5$) | 100 users | 2.68–181.73 |
| Per capita built-up area ($C_6$) | Highest value of all years | 5.12–64.55 |
| Road density ($C_7$) | Highest value of all years | 64.18–758.93 |
| Number of doctors per 10,000 people ($C_8$) | 30 person [50] | 9.38–44.43 |
| Number of beds in hospitals and health centers per 10,000 people ($C_9$) | 60 unit [50] | 15.79–52.25 |
| Number of buses and trams per 10,000 people ($C_{10}$) | Highest value of all years | 0.14–13.55 |
| Number of full-time teachers in universities per 10,000 people ($C_{11}$) | Highest value of all years | 0.58–31.17 |
| Number of books in public libraries per 100 people ($C_{12}$) | 100 unit | 5.00–441.79 |
| Proportion of basic medical insurance ($C_{13}$) | 100% | 0.45–76.32 |
| Gross domestic product ($C_{14}$) | Highest value of all years | 16.30–2301.46 |
| Per capita GDP ($C_{15}$) | Highest value of all years | 4610–107,960 |
| Proportion of tertiary industry in GDP ($C_{16}$) | Highest value of all years | 24.44–79.65 |
| Water consumption per 10,000 yuan GDP ($C_{17}$) | Lowest value of all years | 1.14–48.99 |
| Energy consumption per 10,000 yuan GDP ($C_{18}$) | Lowest value of all years | 0.34–3.34 |
| Comprehensive utilization rate of industrial solid waste ($C_{19}$) | 100% | 12.20–100.00 |
| Remote sensing ecological index ($C_{20}$) | 1 | 0.29–0.64 |
| PM$_{2.5}$ annual average concentration ($C_{21}$) | 35 ug/m$^3$ [51] | 33.32–100.53 |
| Per capita industrial wastewater discharge ($C_{22}$) | Lowest value of all years | 3.87–40.13 |
| Industrial sulfur dioxide emissions per 10,000 people ($C_{23}$) | Lowest value of all years | 10.17–414.95 |
| Harmless treatment rate of domestic garbage ($C_{24}$) | 100% | 28.00–100.00 |
| Domestic sewage treatment rate ($C_{25}$) | 100% | 16.00–100.00 |

After acquiring each subsystem's development value, the combined weight was applied to get each subsystem's weight (Equation (11)).

$$w_n = \frac{w_{AHP(n)} + w_{EW(n)}}{2} \tag{11}$$

where $w_n$ is the $n$th subsystem's combined weight; and $w_{AHP(n)}$ and $w_{EW(n)}$ display the $n$th subsystem's AHP and EM weight, respectively.

To obtain six subsystem's AHP weight, they were first treated as an AHP structure. Then, a judgment matrix was constructed by consulting relevant experts. Next, we utilized Equation (12) to calculate the consistency index (CI$_{AHP}$), the average random consistency index (RI), and the consistency ratio (CR) (Equation (13)) to test the results. Typically, if the CR is less than 0.1, the judgment matrix has a good consistency, indicating that each subsystem's weight is reasonable.

$$CI_{AHP} = \frac{\lambda_{\max} - n}{n - 1} \tag{12}$$

$$CR = \frac{CI_{AHP}}{RI} \tag{13}$$

where $\lambda_{\max}$ denotes the highest real eigenvalue, and $n$ is the number of subsystems. In this study, CI$_{AHP}$ was 0.0254; and CR was 0.0205, which was far less than 0.1, indicating that the judgment matrix has a good consistency.

To obtain six subsystem's entropy weight, we utilized Equations (14)–(18) to calculate their weight.

$$p_{ij} = \frac{N_{ij}}{\sum\limits_{i=1}^{m} N_{ij}}, \ j = 1, 2, \ldots m \tag{14}$$

$$E_j = -k \sum_{i=1}^{m} p_{ij} \ln p_{ij}, \ j = 1, 2, \dots m \tag{15}$$

$$d_j = 1 - E_j, \ j = 1, 2, \dots m \tag{16}$$

$$w_j = \frac{d_j}{\sum\limits_{j=1}^{m} d_j}, \ j = 1, 2, \dots m \tag{17}$$

$$k = \frac{1}{\ln m} \tag{18}$$

where $p_{ij}$ is the ratio of the $j$th subsystem information; $m$ is the sample number, which equals 52; $k$ is a constant; $E_j$ is the information entropy of the $j$th subsystem; $d_j$ is the information entropy redundancy; and $w_j$ is the weight of the $j$th subsystem. After acquiring the six subsystem's AHP and EW weight, each subsystem's combined weight was obtained using Equation (11) (Table 5).

**Table 5.** The weight of each subsystem.

| Subsystem | AHP | EW | Combined Weight |
|---|---|---|---|
| Population | 0.1535 | 0.0387 | 0.0961 |
| Society | 0.2433 | 0.3365 | 0.2899 |
| Economy | 0.3745 | 0.3177 | 0.3461 |
| Resource | 0.0442 | 0.2049 | 0.1245 |
| Ecology | 0.1067 | 0.0516 | 0.0792 |
| Environment | 0.0778 | 0.0506 | 0.0642 |

*2.4. Calculation of CI and CDI*

After calculating the whole JJJ and each subsystem's development degree, the coordinated degree index (CI) was calculated (Equation (19).

$$CI = e^{-\frac{\sigma}{\mu}} \tag{19}$$

where $\sigma$ is the standard deviation; $\mu$ denotes the average value of the whole JJJ and each subsystem's development degree value; and CI represents the coordinated degree value. It ranges from 0 to 1. The higher the CI value, the smaller the development gap of all cities within JJJ. Using the CCD model [10,11], the geometric mean value was computed to represent one subsystem's CDI (Equation (20). CDI also ranges from 0 to 1. A higher CDI indicates that a region's overall development level is high, and the disparities between cities are slight.

$$CDI = \sqrt{CI \times DI} \tag{20}$$

where CI denotes the coordinated degree; and DI denotes the development degree.

## 3. Results
*3.1. The Weight of Each Subsystem*

Table 5 shows the weight of each subsystem. It was found that the order of each subsystem's EM weight was society > economy > resource > ecology > environment > population. As for the AHP method, the order of their weight was economy > society > population > ecology > environment > resource. Regarding the combined weight, the order was economy > society > resource > population > ecology > environment. In general, economy had the highest weight, while the environment had the lowest weight, indicating that it was given the least consideration.

### 3.2. *The Changing Trend of the Whole JJJ and Each Subsystem's DI*

### 3.2.1. The Changing Trend of DI at the City Scale

Figure 2 showed the changing trend of DI at the city scale. The percent value above the bar diagram was the average annual growth rate of each city's DI. ZJK (2.44%) and CD (1.33%) were the top two cities in terms of the average annual growth rate of the population subsystem (Figure 2a), whereas TJ was the only city with a negative rate of change. BJ had the highest average DI value (0.70), followed by TJ (0.63). Regarding the society subsystem, BJ had the highest DI value, especially during 2000–2005, with a sharp increase from 0.58 to 0.86, while HS had the lowest DI value with an average value of 0.22. As for the average annual growth rate, CZ had the highest rate (15.59%), while BJ had the lowest one (4.23%), although all cities had been experiencing a continuous increase in DI values. As for the economy subsystem, BJ and TJ were the top two cities in terms of DI value and annual average growth rate (8.11% and 8.02%), while ZJK and QHD were in the lowest position, with the rate of 1.46% and 1.60%, respectively. Except for ZJK and CD, all the left cities showed a continuously increasing economy subsystem trend.

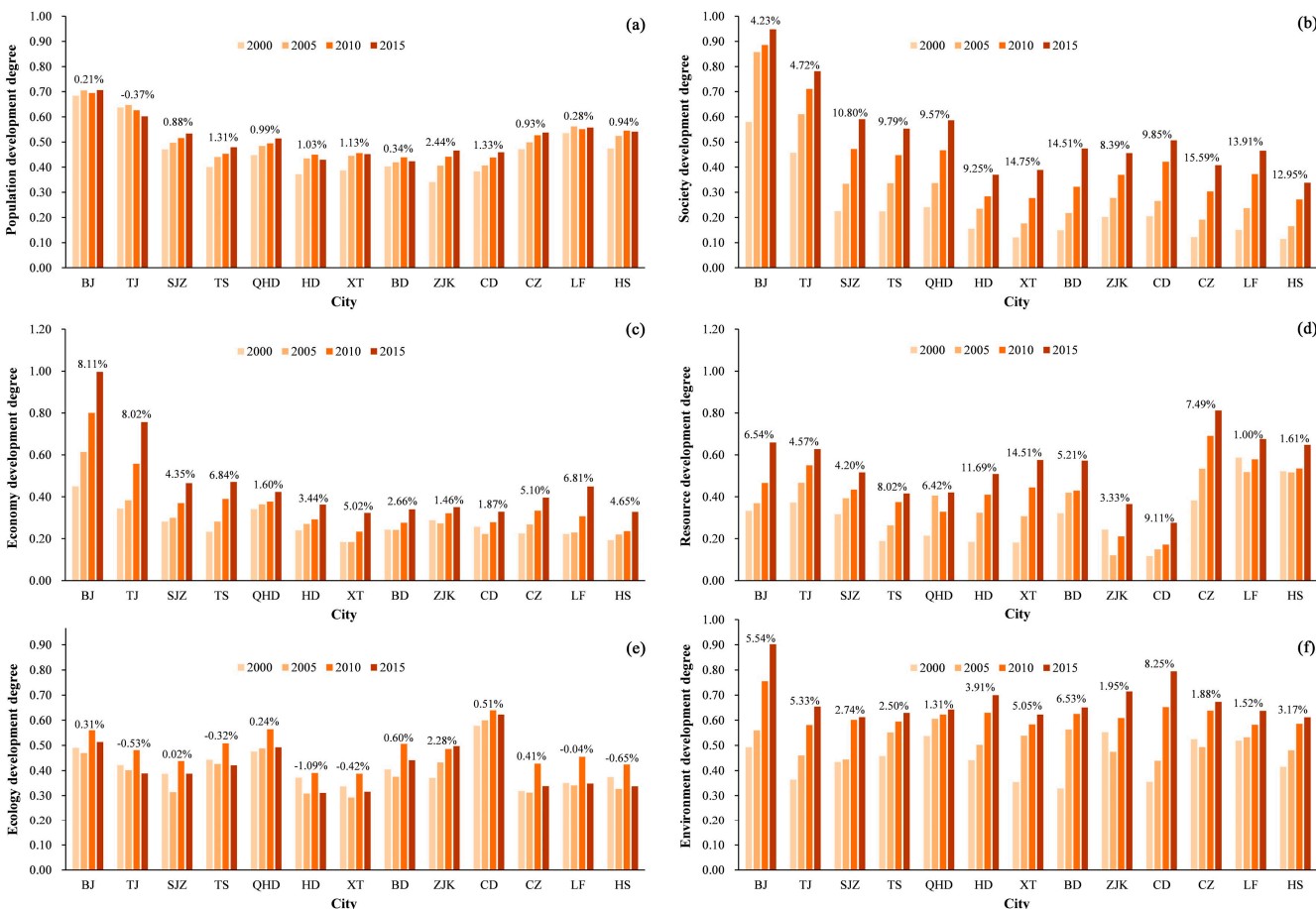

**Figure 2.** The changing trend of the development degree value (DI) at the city scale. (**a**) The DI of the population subsystem; (**b**) The DI of the society subsystem; (**c**) The DI of the economy subsystem; (**d**) The DI of the resource subsystem; (**e**) The DI of the ecology subsystem; (**f**) The DI of the environment subsystem.

Regarding the resource subsystem, CZ had the highest average DI value (0.60), while CD had the lowest (0.18). Its annual average growth rate was highest in XT (14.51%). Regarding the ecology subsystem, all the left cities except for ZJK showed a fluctuating DI trend; Considering the average value of DI, CD had the highest value (0.61), while XT had the lowest one (0.33). Besides, seven cities had a positive annual average rate of change, especially ZJK, that showed the highest annual average growth rate (2.28%). As for the

environment subsystem, BJ had the highest DI value in 2015 (0.90), while HS had the lowest one (0.61), whereas its annual average growth rate was highest in CD (8.25%) and lowest in QHD (1.31%). Generally, BJ achieved the highest average DI value in population, society, economy, and environment, which corresponds to its administrative status; Although CD had the lowest average DI value in resources, it exhibited the highest DI value in ecology, owing to its high vegetation cover.

### 3.2.2. The Changing Trend of DI at the Urban Agglomeration Scale

Figure 3 displayed the changing trend of DI at the urban agglomeration scale. The comprehensive DI value of JJJ showed a continuous increase from 0.30 (2000) to 0.51 (2015), with an annual average growth rate of 4.51%. For each subsystem, the annual average growth rates were the following: society (8.87%), resource (5.23%), economy (4.75%), environment (3.54%), population (0.77%), and ecology (0.47%). By the end of 2015, the environment subsystem had the highest DI value (0.68), followed by resource (0.54), society (0.53), population (0.52), economy (0.46), and ecology (0.46). DI values were divided into five grades, which were poor (0–0.2), fair (0.2–0.4), moderate (0.4–0.6), good (0.6–0.8), and excellent (0.8–1.0). To this end, the DI value grade improved from "fair" to "moderate." The grades of population, society, economy, resource, and ecology were found to be "moderate," and that of the environment was "good" in 2015.

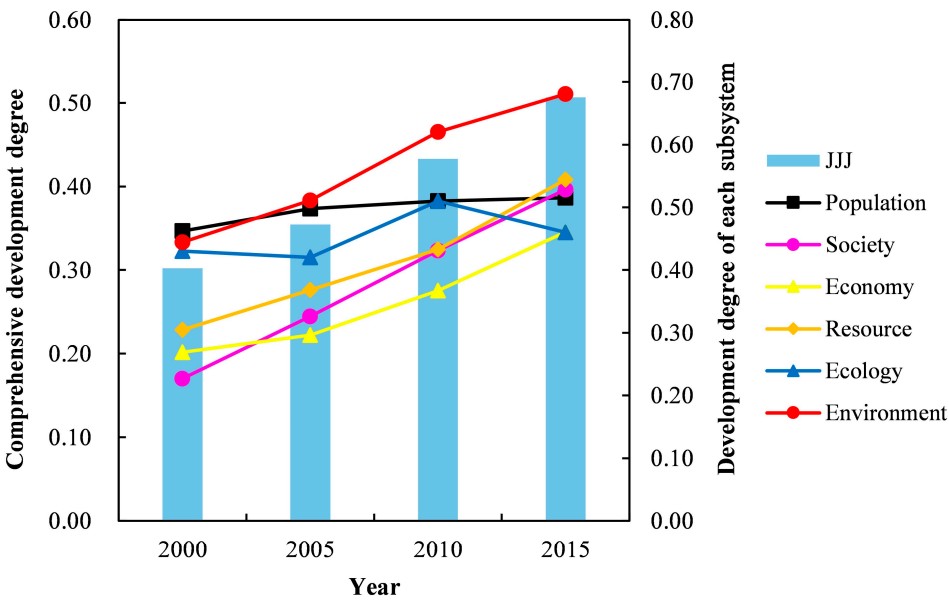

**Figure 3.** The changing trend of DI at the urban agglomeration scale.

### 3.3. The Changing Trend of the Whole JJJ and Each Subsystem's CI Value

Table 6 shows the whole JJJ and each subsystem's CI value. Figure 4a illustrates the changing trend of CI at the urban agglomeration scale. Based on Table 6 and Figure 4, the comprehensive CI of JJJ had a "V" shape. The CI value decreased by −1.74% from 2000 to 2015, indicating the wider gap of comprehensive DI value among cities. Among the six subsystems, population, society, resource, and environment exhibited an increasing trend, while economy and ecology decreased. Their annual average growth rates were the following: society (2.21%), resource (1.34%), population (0.49%), environment (0.35%), ecology (−0.23%), and economy (−0.98%). In general, the CI value of the society subsystem increased significantly from 0.5425 (2000) to 0.7219 (2015), indicating that the gap between cities had been gradually narrowed. However, the CI value of the economy subsystem decreased from 0.7621 (2000) to 0.6505 (2015), indicating that the gap between cities had gradually widened. As with DI, we also divided CI into five grades, equally. The comprehensive CI grade of JJJ was found to be "good" during the study period. The grade of the

population and society subsystems improved from "good" to "excellent" and "moderate" to "good," respectively. The grade of the resource subsystem was not changed ("good" all the time). Furthermore, the grade of both the ecology and environment subsystems was "excellent" during 2000–2015. Although the CI value decreased continuously, the economy subsystem's grade was "good" during the study period.

**Table 6.** The coordinated index (CI) value of the whole Jing-Jin-Ji (JJJ) and each subsystem.

| Subsystem | 2000 | 2005 | 2010 | 2015 |
|---|---|---|---|---|
| Population | 0.7996 | 0.8300 | 0.8557 | 0.8588 |
| Society | 0.5425 | 0.5471 | 0.6566 | 0.7219 |
| Economy | 0.7621 | 0.6884 | 0.6554 | 0.6505 |
| Resource | 0.6356 | 0.6968 | 0.7184 | 0.7632 |
| Ecology | 0.8457 | 0.8069 | 0.8654 | 0.8162 |
| Environment | 0.8401 | 0.9042 | 0.9272 | 0.8843 |
| JJJ | 0.7733 | 0.7404 | 0.7603 | 0.7690 |

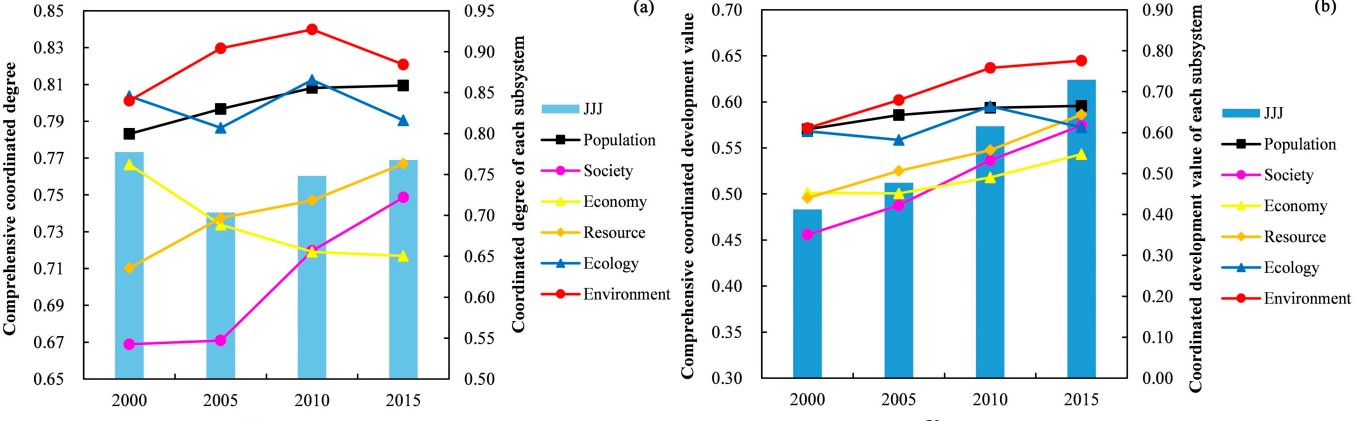

**Figure 4.** The changing trend of CI and CDI at urban agglomeration scale. (**a**) The changing trend of CI; (**b**) The changing trend of CDI.

### 3.4. The Changing Trend of the Whole JJJ and Each Subsystem's CDI

Table 7 displays the CDI value of the whole JJJ and each subsystem. Figure 4b shows the changing trend of CDI at the urban agglomeration. Based on Table 7 and Figure 4b, we found that the comprehensive CDI value of JJJ displayed a continuously increasing trend, from 0.4834 in 2000 to 0.6242 in 2015. The annual average growth rate was 1.94%. The six subsystems' CDI values had a positive annual average change rate; the order was society (5.07%) > resource (3.09%) > environment (1.80%) > ecology (1.39%) > population (0.63%) > ecology (0.11%). At the end of 2015, the order of each subsystem's CDI value was environment (0.7757) > population (0.6653) > resource (0.6444) > society (0.6175) > ecology (0.6127) > economy (0.5475). As with CI and DI, we divided CDI into five grades, equally. Regarding the comprehensive CDI value of JJJ, the grade had improved from "moderate" in 2000 to "good" in 2015. As for the six subsystems, the grade of population was "good" during 2000–2015; society's grade had improved from "fair" to "good"; economy's grade was "moderate" all the time; resource's grade had improved from "moderate" to "good"; ecology's grade had firstly decreased from "good" to "moderate", then improved to "good". In general, combined with CI and DI, we found that the DI and CI value of the economy subsystem displayed a continuously increasing and decreasing trend, respectively; however, the CDI value showed a continuously increasing one, indicating that the gap of all cities' DI growth rate had widened.

**Table 7.** The coordinated development index (CDI) value of the whole JJJ and each subsystem.

| Subsystem | 2000 | 2005 | 2010 | 2015 |
|---|---|---|---|---|
| Population | 0.6078 | 0.6427 | 0.6607 | 0.6653 |
| Society | 0.3507 | 0.4224 | 0.5321 | 0.6175 |
| Economy | 0.4529 | 0.4516 | 0.4905 | 0.5475 |
| Resource | 0.4402 | 0.5065 | 0.5575 | 0.6444 |
| Ecology | 0.6030 | 0.5822 | 0.6643 | 0.6127 |
| Environment | 0.6110 | 0.6798 | 0.7584 | 0.7757 |
| JJJ | 0.4834 | 0.5123 | 0.5737 | 0.6242 |

## 4. Discussion

### 4.1. Weight Assignment with Existing Studies

In our study, each subsystem's weight was assigned by combining AHP and EM. We found that the economy subsystem had the highest weight, followed by society, resource, population, ecology, and environment. The economy was one of the most critical subsystems. Previous studies had paid much attention to building economic subsystems. Li and Yi treated the economy as one of the necessary subsystems [3]; Liu et al. constructed an indicator system from the aspects of the economy, society, and environment, with the order of their weight being economy > society > environment [36], which was consistent with our result. Fang et al. studied the green development under the population-resource-environment-development-satisfaction perspective and found that resource had a higher weight than the population subsystem [43]. Besides, several studies assigned the same weight to different subsystems. For instance, Wang et al. analyzed the energy, economy, and environment subsystems by giving the same weight to each of them [34]; Guan et al. also gave the same weight to the urban economy-resource-environment system [28]. However, we thought that different subsystems should have different weights. Former studies thought that the coordinated development should focus on the synchronous development of different subsystems; hence, they gave the same weight to different subsystems. However, based on our definition, we focused on improving the DI of different subsystems and narrowing the DI gap between cities. We believe that the improved DI of different subsystems is ideal for seeking a synchronous development of different subsystems. Since the implementation of the reform and opening-up policy in China, the government puts much emphasis on the economy subsystem. Thus, based on our findings, assigning different weights to different subsystems would be more appropriate.

### 4.2. Analysis of DI, CI, and CDI and Their Implications

Based on the DI value at the city level, TJ's population DI showed a negative annual average growth rate (−0.37%) due to its low population quantity and population score, from a value of 0.52 in 2000 to 0.26 in 2015. However, although TJ showed a decreasing trend, it had been ranked second all the time. As one of China's municipalities, TJ was the Bohai Sea area's economic center according to the Master Plan of Tianjin (2005–2020). Regarding the society subsystem, the DI value of BJ had sharply increased from 0.58 in 2000 to 0.86 in 2005, mainly because of the sharp increase of the indicator $C_5$. During 2000–2005, the original value of $C_5$ had increased from 25.46 (users) to 94.92 (users). Regarding the economy subsystem, the rankings of LF had improved from eleventh (2000) to fifth (2015). Since the Beijing-Tianjin-Hebei Collaborative Development Strategy promotion in 2014, LF has played an increasingly important role in connecting BJ and TJ. Regarding the resource subsystem, in 2015, CZ had the highest DI value owing to the high score of $C_{17}$; the original value was 1.14 (tons), indicating that CZ had a high utilization efficiency of water resources. CD had the lowest DI value in 2015 because of its low water, energy, and industrial solid waste utilization efficiency. For BJ, its ranking improved from fifth in 2000 to third in 2015 due to its high energy utilization efficiency; however, we found that its water utilization efficiency was very low in 2015 (0.14), and needed to increase in the future. Zhu et al. also reported that BJ's resource utilization's overall level was

relatively insufficient [47]. Regarding the ecology subsystem, CD had the highest DI value all the time, which had a good relationship with its high vegetation cover. Based on the statistical data from Chengde Forestry and Grassland Bureau, CD had approximately 2.34 million hectares of forest land, accounting for 32% of the total JJJ; Besides, in 2020, the forest cover rate of CD was 59.41%, which was higher than the national level by 36%; the grassland comprehensive vegetation coverage rate was 73.60%. The DI value of ZJK displayed a continuously increasing trend, owing to a series of projects implemented by the government, such as "Returning Farmland to Forest (grass) Project", "Three-North Shelter Forest Program", "Beijing-Hebei Ecological Water Resources Protection Forest Project", etc. To further understand the ecological change situation, based on the normalized ecological time difference index (NETDI) in our previous work [25], all cities' average NETDI values were calculated based on Equation (21).

$$NETDI = \frac{EQ_{T_{end}} - EQ_{T_{start}}}{EQ_{T_{end}} + EQ_{T_{start}}} \tag{21}$$

where $EQ_{Tstart}$ and $EQ_{Tend}$ denote the ecological quality in 2001 and 2015; NETDI ranges from $-1$ to 1—the higher the value, the higher the improvement of ecological quality, and vice versa.

Table 8 shows the average NETDI value of each city. It could be found that BJ, QHD, BD, ZJK, CD, and CZ had a positive value, indicating that these cities' ecological quality had improved. For BJ, similar results were also reported in the Beijing Municipal Ecological Remote Sensing Annual Report (2018). For ZJK, as we have mentioned above, series projects had been mainly implemented in this city. However, TJ, SJZ, TS, HD, XT, LF, and HS had a negative value, indicating that these cities' ecological quality had slightly declined. Wang et al. found similar results: the ecological quality of CD, BJ, QHD, and BD was higher than JJJ's average level, while LF, TJ, and CZ were lower than JJJ's average level [52].

**Table 8.** The average normalized ecological time difference index (NETDI) value of each city.

| City | NETDI | City | NETDI |
|------|-------|------|-------|
| BJ | 0.0252 | BD | 0.0438 |
| TJ | −0.0443 | ZJK | 0.1633 |
| SJZ | −0.0025 | CD | 0.0418 |
| TS | −0.0275 | CZ | 0.0450 |
| QHD | 0.0160 | LF | −0.0029 |
| HD | −0.0962 | HS | −0.0511 |
| XT | −0.0339 | | |

Regarding the environment subsystem, all cities' DI values had increased during 2000–2015 due to the sharp improvement of the environmental governance level, which had increased from 0.56 to 0.96. The population subsystem's growth rate gradually slowed down at the urban agglomeration scale due to its low development value in population quantity. Zhu et al. calculated all cities' population carrying capacity and found that all JJJ cities need population regulation (such as population structure improvement and optimization of population spatial distribution pattern) [47].

Based on the CI value at the urban agglomeration scale, population, ecology, and environment had a relatively high CI value, indicating that their DI gap between cities was not high. Regarding the society subsystem, the CI value had increased a lot since 2005, indicating that the basic public service level and infrastructure construction level had improved. To achieve an equalization of basic public services, the Chinese central government decided to steadily improve public services equalization during the 13th 5-Year Plan period (2016–2020) [53]. Regarding the economy subsystem, its CI value displayed a continuously decreasing trend, although all cities' economy DI had improved. However, the economic gap between cities had widened. Zhao et al. also found that the economic disparity in JJJ had widened [54]. Table 9 shows the CI value of all indicators.

**Table 9.** The CI value of all indicators during 2000–2015.

| Indicator | CI | | | | Indicator | CI | | | |
|---|---|---|---|---|---|---|---|---|---|
| | **2000** | **2005** | **2010** | **2015** | | **2000** | **2005** | **2010** | **2015** |
| $C_1$ | 0.5389 | 0.5510 | 0.5091 | 0.4544 | $C_{14}$ | 0.4016 | 0.3185 | 0.3098 | 0.2880 |
| $C_2$ | 0.8567 | 0.8741 | 0.8760 | 0.8502 | $C_{15}$ | 0.5682 | 0.5545 | 0.5665 | 0.5725 |
| $C_3$ | 0.5844 | 0.7404 | 0.8015 | 0.8613 | $C_{16}$ | 0.7829 | 0.7343 | 0.7346 | 0.7737 |
| $C_4$ | 0.9131 | 0.9058 | 0.9106 | 0.8786 | $C_{17}$ | 0.4684 | 0.4729 | 0.4830 | 0.4761 |
| $C_5$ | 0.5413 | 0.5929 | 0.7980 | 0.8739 | $C_{18}$ | 0.5030 | 0.6968 | 0.6198 | 0.6584 |
| $C_6$ | 0.4885 | 0.4647 | 0.4873 | 0.5157 | $C_{19}$ | 0.6503 | 0.6727 | 0.6952 | 0.7625 |
| $C_7$ | 0.5964 | 0.6354 | 0.6166 | 0.6306 | $C_{20}$ | 0.8457 | 0.8069 | 0.8654 | 0.8162 |
| $C_8$ | 0.6808 | 0.6882 | 0.7807 | 0.8435 | $C_{21}$ | 0.7366 | 0.7035 | 0.6983 | 0.7253 |
| $C_9$ | 0.6814 | 0.6890 | 0.8475 | 0.8993 | $C_{22}$ | 0.5482 | 0.6659 | 0.5312 | 0.5392 |
| $C_{10}$ | 0.2682 | 0.2510 | 0.4739 | 0.5010 | $C_{23}$ | 0.5055 | 0.4829 | 0.4358 | 0.2506 |
| $C_{11}$ | 0.2647 | 0.3787 | 0.4497 | 0.4390 | $C_{24}$ | 0.7072 | 0.8524 | 0.9451 | 0.9672 |
| $C_{12}$ | 0.3432 | 0.3733 | 0.4051 | 0.4651 | $C_{25}$ | 0.6902 | 0.7898 | 0.9517 | 0.9353 |
| $C_{13}$ | 0.4047 | 0.4260 | 0.3744 | 0.5156 | | | | | |

Based on Table 9, it could be found that the CI values of nine indicators ($C_1$, $C_2$, $C_4$, $C_{14}$, $C_{16}$, $C_{20}$, $C_{21}$, $C_{22}$, and $C_{23}$) exhibited a decreasing trend during 2000–2015. $C_{23}$ had the lowest rate (−3.36%), indicating that the gap of this indicator between cities had widened from the aspects of the annual average growth rate. Furthermore, $C_{21}$, $C_{22}$, and $C_{23}$ all belonged to the environmental pollution attribute. Regarding the indicators with a positive rate, $C_{10}$ had the highest value (5.79%). Besides, all society subsystem indicators displayed a positive change rate, indicating that the gap of society development level between cities had narrowed. By the end of 2015, the CI value of $C_1$, $C_{11}$, $C_{12}$, $C_{14}$, $C_{17}$, and $C_{23}$ was lower than 0.5. Based on the CDI value at the urban agglomeration scale, the whole JJJ and six subsystems all displayed an increasing trend, indicating that the coordinated development level had improved, which was similar to Zhu's findings [47]. However, based on Table 10, it could be found that ten indicators' CDI ($C_1$, $C_6$, $C_{10}$, $C_{11}$, $C_{12}$, $C_{13}$, $C_{14}$, $C_{17}$, $C_{22}$, and $C_{23}$) was lower than 0.5 at all times, and these indicators mainly belonged to the basic public service and environmental pollution aspects, indicating that the government still needs to work hard to improve both aspects.

**Table 10.** The CDI value of all indicators during 2000–2015.

| Indicator | CDI | | | | Indicator | CDI | | | |
|---|---|---|---|---|---|---|---|---|---|
| | **2000** | **2005** | **2010** | **2015** | | **2000** | **2005** | **2010** | **2015** |
| $C_1$ | 0.3779 | 0.3715 | 0.3104 | 0.2591 | $C_{14}$ | 0.1128 | 0.1490 | 0.2133 | 0.2596 |
| $C_2$ | 0.7774 | 0.8105 | 0.8236 | 0.8026 | $C_{15}$ | 0.2296 | 0.3104 | 0.4330 | 0.5218 |
| $C_3$ | 0.5233 | 0.6720 | 0.7420 | 0.8173 | $C_{16}$ | 0.6105 | 0.5879 | 0.6031 | 0.6622 |
| $C_4$ | 0.6983 | 0.7135 | 0.7457 | 0.7450 | $C_{17}$ | 0.1607 | 0.2358 | 0.3279 | 0.4013 |
| $C_5$ | 0.2253 | 0.4678 | 0.7344 | 0.8716 | $C_{18}$ | 0.3436 | 0.4214 | 0.4368 | 0.5504 |
| $C_6$ | 0.3438 | 0.3905 | 0.4210 | 0.4650 | $C_{19}$ | 0.6374 | 0.7019 | 0.7304 | 0.7973 |
| $C_7$ | 0.3549 | 0.4568 | 0.4659 | 0.5473 | $C_{20}$ | 0.6030 | 0.5822 | 0.6643 | 0.6127 |
| $C_8$ | 0.6140 | 0.5828 | 0.6971 | 0.7992 | $C_{21}$ | 0.6137 | 0.5939 | 0.5886 | 0.6599 |
| $C_9$ | 0.5447 | 0.5515 | 0.7155 | 0.8079 | $C_{22}$ | 0.4537 | 0.4161 | 0.4116 | 0.4833 |
| $C_{10}$ | 0.1915 | 0.2214 | 0.3463 | 0.3754 | $C_{23}$ | 0.2222 | 0.1957 | 0.2289 | 0.2140 |
| $C_{11}$ | 0.2087 | 0.3405 | 0.3974 | 0.4026 | $C_{24}$ | 0.7303 | 0.8584 | 0.9614 | 0.9748 |
| $C_{12}$ | 0.3130 | 0.3394 | 0.3732 | 0.4215 | $C_{25}$ | 0.5032 | 0.6994 | 0.9149 | 0.9319 |
| $C_{13}$ | 0.1436 | 0.2232 | 0.2847 | 0.3909 | | | | | |

*4.3. Limitations and Further Study*

The CDI of the JJJ urban agglomeration during 2000–2015 has been investigated. However, this study still has some limitations. Firstly, the indicator system is not imperfect. Although we have carefully selected representative indicators to evaluate the CDI, we

had to ignore some aspects; secondly, as we have discussed above, the six subsystems' weight was determined by the combination of AHP and EM. However, weight assignment involves numerous aspects; in the future, more studies could be done to combine and compare newly developed weighting methods to assign the different subsystems a more appropriate weight. Finally, in our study, we only calculated the CDI of JJJ in the past years; although these results could provide some suggestions for relevant policymakers, we still need to predict the future state with some models, like the system dynamic model.

## 5. Conclusions

This study used statistical data and non-statistical data to evaluate JJJ's DI, CI, and CDI at the city and urban agglomeration scales during 2000–2015. The main conclusions were as follows: (1) The economy subsystem had the highest weight; (2) At the city scale, BJ had the highest comprehensive DI value, followed by TJ; at the urban agglomeration scale, the comprehensive DI value of JJJ had increased from 0.30 in 2000 to 0.51 in 2015, with the improvement of grade from "fair" to "moderate"; (3) The comprehensive CI value of JJJ displayed a "V" shape, with an annual average growth rate of $-0.04\%$; among the different subsystems, the CI value of the economy subsystem showed a continuous decrease from 0.7621 (2000) to 0.6505 (2015); (4) The comprehensive CDI value of JJJ increased from 0.39 to 0.59, with the change of its grade from "fair" to "moderate" during the study period.

**Author Contributions:** Conceptualization, J.J. and S.W.; methodology, J.J.; software, J.J..; validation, Y.Z., W.L. and L.W.; formal analysis, S.W.; investigation, J.J.; resources, W.L; data curation, Y.Z.; writing—original draft preparation, J.J; writing—review and editing, S.W.; visualization, Y.Z.; supervision, W.L.; project administration, W.L.; funding acquisition, W.L. All authors have read and agreed to the published version of the manuscript.

**Funding:** This research was funded by the National Key Research and Development Program of China, grant number 2017YFB0503805.

**Institutional Review Board Statement:** Not applicable.

**Informed Consent Statement:** Not applicable.

**Data Availability Statement:** The data presented in this study are available on request from the first author.

**Acknowledgments:** We thank the national and local statistics bureau and the United States Geological Survey (USGS) for supplying the socioeconomic and remote sensing datasets. We thank Eshetu Shifaw for his kind help in English language editing.

**Conflicts of Interest:** The authors declare no conflict of interest.

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
