# Peer review of "Spatiotemporal Change and Coordinated Development Analysis of “Population-Society-Economy-Resource-Ecology-Environment” in the Jing-Jin-Ji Urban Agglomeration from 2000 to 2015"

_sustainability, doi:10.3390/su13074075_

Round 1

Reviewer 1 Report

The manuscript (ms) proposed a new index to assess the coordinated development combining statistical and remote sensing data. The index was applied to analyze the coordinated development degree of the Jing-Jin-Ji (JJJ) area from 2000 to 2015. Overall, it is a well-prepared paper with detailed methods and experiments. However, my main concern is the novelty of this paper. The authors have published several similar journal articles before. For example:

Ji, J., Wang, S., Zhou, Y., Liu, W., & Wang, L. (2020). Studying the eco-environmental quality variations of Jing-Jin-Ji urban agglomeration and its driving factors in different ecosystem service regions from 2001 to 2015. IEEE Access, 8, 154940-154952.

Ji, J., Wang, S., Zhou, Y., Liu, W., & Wang, L. (2020). Spatiotemporal change and landscape pattern variation of eco-environmental quality in Jing-Jin-Ji Urban Agglomeration From 2001 to 2015. IEEE Access, 8, 125534-125548.

The processing flow of MODIS data is very similar. The study area and cities as well as many figures and results are exactly same with the above papers. The authors should notice the work submitted in a manuscript must be original and not published elsewhere.

General remarks:

Introduction:

The literature review provided at the beginning of the introduction needs to be reorganized. It looks like a long list of article titles/authors but lacks organization. The literatures should be grouped and reviewed according to the method, application, or case. Also, the authors should discuss the advantages and limitations of the previous works. 

Materials and Methods:

Figure 1 failed to display the location of the study area in the country. The author should insert a mini-map to show the location of the JJJ area in China. Also, the image quality of this figure is quite poor. Please use a high-resolution image. 

There are several points the authors need to notice when selecting indicators. First of all, I do not think the “population quality” is measurable. You may use the per-capita years of education to measure “education years” or “education level”, but should be really careful to use the word “population quality”. Second, please provide the details of the data source of each indicator. For the MODIS data, how many images have you used? What about the cloud coverage issue? Besides, please specify the basic characteristics, such as spatial/temporal resolution, data availability, the difference between MOD09AI and MOD11A2. For Table 2, I suggest presenting the value range of each indicator.

Results and discussion:

Please improve the image quality of all figures. For Figure 8, could you please annotate the city name/location on the map?

These two sections present the result and discussion of historical trends of each subsystem. However, the experiment failed to assess the result using the other reference data set. Have you considered comparing the development degree with other historical data published by the government to support the conclusion? 

Detailed remarks:

Line 17: Please specify the full name of “JJJ” before the first use.

Line 92: Can you specify which existing studies?

Line 102: Any references?

Line 105: It is “Google Earth Engine”.

Line 195: Please specify the studies.

Line 226: It is “Google Earth Engine platform”

Line 290: “CI” has been defined at line 272.

Author Response

Dear Professor,

Our response letter and the revised manuscript had been uploaded as an attachment.

Reviewer 2 Report

Dear authors,

there is no duplicate in this study, good structure. However, there are some minor mistakes, please revise them:

Line 12: the adjective "regional' is modifying coordinated instead of a noun or pronoun. Use an adverb to modify a verb, adjective, or other adverbs. Change to REGIONALLY.

Line 14: delete 'in this'; delete 'we', rewrite this sentence: see bellow: 
This study integrated remote sensing technology and traditional statistical data to evaluate the development degree from six subsystems.

Line 94: it appears that 'In fact, may be unnecessary in this sentence. consider removing it.

Equations 8 to equation 18 are not mentioned in the relevant content.

In all the other parts, writing is very well. Please revised those minor mistakes before considering for the publication process.

Author Response

Dear Professor,

Our response letter and the revised manuscript had been uploaded as an attachment. Please see the attachment. Many thanks for your constructive suggestions.

Reviewer 3 Report

The topic of this manuscript is interesting. The coordinated development in a metropolitan region is investigated using multiple data. Many aspects of this manuscript should be improved.

1. The English really needs to improve. Many sentences do not write clearly, which makes me hard to understand and follow.

2. The title would be better to change to a shorter one. It is not necessary to use the term “Population-Society-Economy-Resource-Ecology-Environment”

3. You defined coordinated development index (CDI) in Sections 1 and 2, but why not use the term, CDI, in the following sections? I was wondering Which table is CDI in section 3? The Table 5 or Table 6?

4. Equations (2) - (7) are important for the data processing in this manuscript. Please add the reference for these equations.

5. In some parts I did not follow well. For example, How did you calculate the omega (weight of subsystem) in equation (10)?

6. In section 2.5. For the coordinated development index (CDI), is it defined by yourself? If so, why do you take root square of it? If not, please list the reference.

7. Section 3.2 needs shrink. Lots of detailed descriptions about the figures are not necessary and redundant.

8. There are 7 figures in Section 3 that keep the exact same format. If it is possible, would you please update or combine the figures to show the results in a more concise and nice way?

9. The results are not very strong. Most of them are just summaries. I suggest the authors think about it again and reorganize it.

10. I believe the 3.1.1-3.1.7 should be 3.2.1-3.2.7.

Author Response

(The authors gave the same response as above.)

Round 2

Reviewer 1 Report

The manuscript was improved and revised accordingly. I think the paper is acceptable to be published.

Author Response

Dear reviewer,

We would like to express our sincere thanks for your constructive and positive comments on our manuscript. Thanks for your sincere help again.

Best wishes,

Jianwan Ji et al.

Reviewer 3 Report

The author did some improvement for this manuscript, such as the previous figures are updated. However, the improvement far not says implement a major revision.
All the equations in this paper need careful check. The equations in a paper should let readers able to understand your idea and able to reproduce your results, but many places in these equations are not clear enough. For example, What is LST in equation (4)? What is m in equation (18)?
The word selection is still a problem, although the author claims did a language polish.
Could you list the source of the ideal values in Table 4? Just curious how did you get the ideal values?

Author Response

Dear reviewer,

We would like to express our sincere thanks for your constructive and positive comments. We have carefully handed in all the comments and revised our manuscript. Thanks for your suggestions again.

Best wishes,

Jianwan Ji et al.
